# ReMI: A Dataset for Reasoning with Multiple Images

**Mehran Kazemi**[1], **Nishanth Dikkala**[2], **Ankit Anand**[1], **Petar Devic**[3],
**Ishita Dasgupta**[1], **Fangyu Liu**[1], **Bahare Fatemi**[2], **Pranjal Awasthi**[2],
**Dee Guo**[2], **Sreenivas Gollapudi**[2], **Ahmed Qureshi**[3]
[1]Google DeepMind, [2]Google Research, [3] Google

## Abstract

With the continuous advancement of large language models (LLMs), it is essential to create new benchmarks to effectively evaluate their expanding capabilities and identify areas for improvement. This work focuses on multi-image reasoning, an emerging capability in state-of-the-art LLMs. We introduce ReMI, a dataset designed to assess LLMs' ability to **Re**ason with **M**ultiple **I**mages. This dataset encompasses a diverse range of tasks, spanning various reasoning domains such as math, physics, logic, code, table/chart understanding, and spatial and temporal reasoning. It also covers a broad spectrum of characteristics found in multi-image reasoning scenarios. We have benchmarked several cutting-edge LLMs using ReMI and found a substantial gap between their performance and human-level proficiency. This highlights the challenges in multi-image reasoning and the need for further research. Our analysis also reveals the strengths and weaknesses of different models, shedding light on the types of reasoning that are currently attainable and areas where future models require improvement. To foster further research in this area, we are open-sourcing ReMI: https://huggingface.co/datasets/mehrankazemi/ReMI.

## 1  Introduction

Large Language Models (LLMs) have demonstrated an extraordinary evolution, not only in their output quality but also in their burgeoning capabilities. A significant direction of development has been models' ability to perform increasingly general forms of reasoning that were previously not possible. The emergence of these novel capabilities necessitates the development of robust evaluation benchmarks and metrics to measure and enhance model performance in these specific areas.

The ability of LLMs to reason over text has improved in leaps and bounds, and has been studied extensively [25, 49, 37]. More recent developments in multi-modal models has opened up a new space of reasoning problems, moving toward the capability to reason across multiple, potentially disparate, sources of information presented in various formats [38, 46, 1, 5, 24, 19, 28]. This multi-modal reasoning capability has numerous applications, from complex problem-solving to information synthesis. In this paper, we focus on a specific aspect of this capability: multi-image reasoning. A large portion of the current benchmarks for multi-modal evaluation are based on a single image [33, 34, 32, 20]. We address the lack of dedicated evaluation frameworks in this domain by introducing a comprehensive benchmark designed to specifically assess and improve this skill in LLMs. We focus specifically on reasoning problems where besides visual understanding, one needs to find a step-by-step solution to a problem. This process often involves combining information across text and multiple images – a skill that is currently not extensively evaluated in existing benchmarks. This contribution aims to catalyze progress in multi-image reasoning, ultimately enabling LLMs to better navigate and extract insights from the increasingly complex information landscape of our digital world.

38th Conference on Neural Information Processing Systems (NeurIPS 2024) Track on Datasets and Benchmarks.

We introduce ReMI, a new benchmark designed for **Re**asoning with **M**ultiple **I**mages. Our goal is to cover a broad spectrum of domains where integrating information across multiple modalities is necessary, as well as various key properties unique to multi-image reasoning. To achieve this, we developed 13 tasks that span a range of domains and properties. The domains covered in ReMI include algebra, calculus, geometry, graph theory, physics, temporal and spatial/maps reasoning, tabular and chart understanding, coding, and logic. The properties covered by ReMI include sequential vs set consumption of image information, problems that require reasoning over images demonstrating a similar concept (e.g., two charts) or different concepts (e.g., geometry shape and a table), images that are interleaved with the text or not, and the number of separate images provided as input. Our tasks require reasoning over at least two and up to six images. Table 1 outlines the tasks, domains and properties. Our images comprise a variety of heterogeneous image types including charts, tables, equations, emojis, graphs, shapes, maps, clocks, objects, LaTeX diagrams, functions, etc.

We evaluate state-of-the-art LLMs on ReMI and compare their performance to humans, showing that model performances remain substantially behind human performance (see Fig 1). Interestingly, our results also reveal that models may perform better when multiple images are fed to them separately as opposed to all in one image; this is especially true in the case where the images are interleaved with the question text. A detailed failure analysis reveals model shortcomings that can guide future improvement efforts.

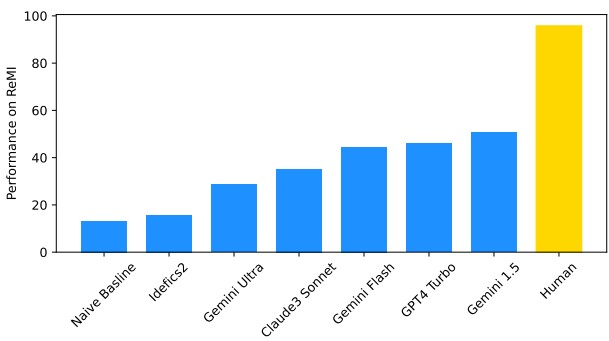

Figure 1: Model performances on ReMI.

## 2   Related Work

**Vision-language foundation models.** In our work, we focus on vision language generation models, i.e. models that produce open-ended text conditioned on text and images. Frozen [47] and Flamingo [3] first transformed LLMs into vision-language models by adding a vision transformer tower and training cross/self-attention layers to enable LLMs to perceive visual information. Subsequently, a large volume of research emerged focusing on the approach of stitching a pretrained visual encoder (usually vision transformer) to a pretrained langauge model. PaLI [9], BLIP [27], LLaVA [31], OpenFlamingo [6], PaLIGemma [8] all follow similar techniques. The latest closed-source frontier models such as GPT-4 [1], Gemini [46] and Claude 3 [5] all have vision input support and are also reported to be the best performing models across popular vision-language reasoning benchmarks [33]. These frontier models are able to condition fairly arbitrarily on sequences of interleaved image and text. However, most vision-language benchmarks test models' performance on a single image-text pair; the focus of this paper is to take a step toward evaluating more flexible vision-language abilities.

**Reasoning Benchmarks.** Reasoning has been a core area of interest for NLP systems. The initial benchmarks focused on 'simpler' reasoning tasks which largely involve language understanding (e.g. SuperGLUE [48], HellaSwag [51], Lambada [36]). With LLMs making remarkable strides in recent years, a plethora of benchmarks requiring much stronger reasoning abilities have emerged. Some of these like MMLU [15] and ARC [10] focus on science questions. MATH [16], GSM8K [11] and MGSM [40] focus on mathematical problem solving. There is also a line of works [45, 39, 21] which construct semi-synthetic benchmarks to evaluate the logical deductive reasoning abilities of LLMs. In addition, the BIG-Bench [42] suite of tasks contains many which focus on reasoning.

**Vision-language reasoning benchmarks.** Some recent benchmarks like [50, 33, 20] present reasoning problems that require conditioning on images; however, they predominantly require only a single image, and do not directly measure how well the model can integrate information across different images. Cross-image reasoning benchmarks exist but are restricted to the entailment task or focus on limited number of domains. NLVR [43] creates pairs of images composed of synthetic 2D objects and the task is identifying whether the caption is entailed from the image. MLLM-CompBench [23] creates pairs of images and asks comparative questions about them. NLVR2 [44] extends NLVR by

replacing synthetic images with pairs of images sampled from MS COCO [29]. MaRVL [30] expands a similar idea to multi-cultural and multilingual scenarios and only focuses on the natural image domain. MileBench [41] develops a multi-modal long-context benchmark. SEED-Bench-2 [26] proposes a hierarchy of different vision-language datasets including multi-image datasets composed of frames extracted from videos. BLINK [13] is a collection of 14 visual perception tasks where some of the tasks involve multiple images, e.g. visual similarity and multi-view reasoning. None of these mentioned benchmarks aim to test vision-language models for *complex reasoning* in *multi-image* scenarios. We aim to propose a holistic benchmark covering a wide range of visual information in the world and focuses on complex reasoning of multi-images.

## 3 The ReMI Dataset

Multi-image reasoning can arise in many domains and the problems involving reasoning over multiple images may differ in some key properties. We aim to create a benchmark that exhibits many domains and covers those key properties as much as possible. To this end, we included 13 tasks in our benchmark that covers the following domains: *Algebra*, *Calculus*, *Geometry*, *Tabular Reasoning*, *Time Arithmetic*, *Logic*, *Physics*, *Spatial Reasoning*, *Graph Theory*, *Charts*, *Maps*, and *Coding*. We also identified the following key properties specific to multi-image reasoning and aimed for having tasks that provide a good coverage of them:

Table 1: The properties of the tasks in our benchmark.

| Task Name | Reasoning Domain(s) | Sequence or Set | Same/Diff. Concept | Interleaved | Max #Images |
|---|---|---|---|---|---|
| EmojiAlgebra | Algebra | Seq | Same | Yes | 6 |
| FuncRead | Calculus | Mix | Same | Yes | 3 |
| GeomShapes | Geometry | Seq | Same | No | 2 |
| GeomCost | Geometry, Tabular | Seq | Diff | Yes | 2 |
| Collisions | Physics | Set | Same | No | 2 |
| Clocks | Time Arithmetic | Set | Same | No | 2 |
| Schedule | Time, Tabular | Seq | Diff | Yes | 2 |
| Charts | Charts | Set | Same | No | 2 |
| CodeEdit | Code | Seq | Same | Yes | 2 |
| Isomorphism | Graph Theory | Set | Same | Yes | 2 |
| Maps | Spatial, Maps | Mix | Same | Yes | 4 |
| RefCOCO | Spatial | Seq | Diff | No | 2 |
| IQ | Logic | Mix | Same | Yes | 5 |

- **Sequential vs Set:** In some tasks, the provided images have to be consumed in a sequence (e.g., computing a quantity from one image and then using that quantity in the second image), whereas in some other tasks, the provided images constitute a set. When more than two images are provided, they may be grouped into subsets that have to be consumed sequentially.

- **Same vs Different Concept:** In some multi-image reasoning problems, the provided images all correspond to the same concept (e.g., all of them are charts, or function graphs) whereas in some other problems, the provided images may correspond to different concepts (e.g., one image might be a geometry shape, and the other might be a table).

- **Interleaving:** For all our tasks, we can either provide all the images first and then ask a question about them, or the images can be interleaved with the question task when they are referred to. To enable experimenting for both settings, we make a subset of the tasks interleaved while for the others we provide the image at the beginning of the prompt.

- **Number of images:** In some tasks, a variable number of images may be provided as input.

Solving our tasks requires parsing and understanding the information in the images and text of the question provided as input, which is often followed by the model having to reason using this information to arrive at the correct answer. We provide a brief description of each task below and a more detailed description in the Appendix. In Figure 2, we illustrate a sample from each of the tasks in ReMI. Moreover, in Table 1, we specify the domain and properties for each of the tasks in ReMI.

(1) EmojiAlgebra: Solve a system of linear equations involving digits and emojis. Each image contains an equation or the final expression to be computed. (2) FuncRead: Given multiple function graphs in separate images, answer questions about them. (3) GeomShapes: Given two shapes (in two different images) with a common property, compute a missing value of one of the shapes. (4) GeomCost: Given the shape of an object (in one image) on which an operation is to be done and a table of various costs (in a different image), compute the total cost of the operation. (5) Collisions: Given the before and after snapshots of two objects colliding (each in a separate image), answer questions about their state. (6) Clocks: Given two clocks with different designs (each in a separate

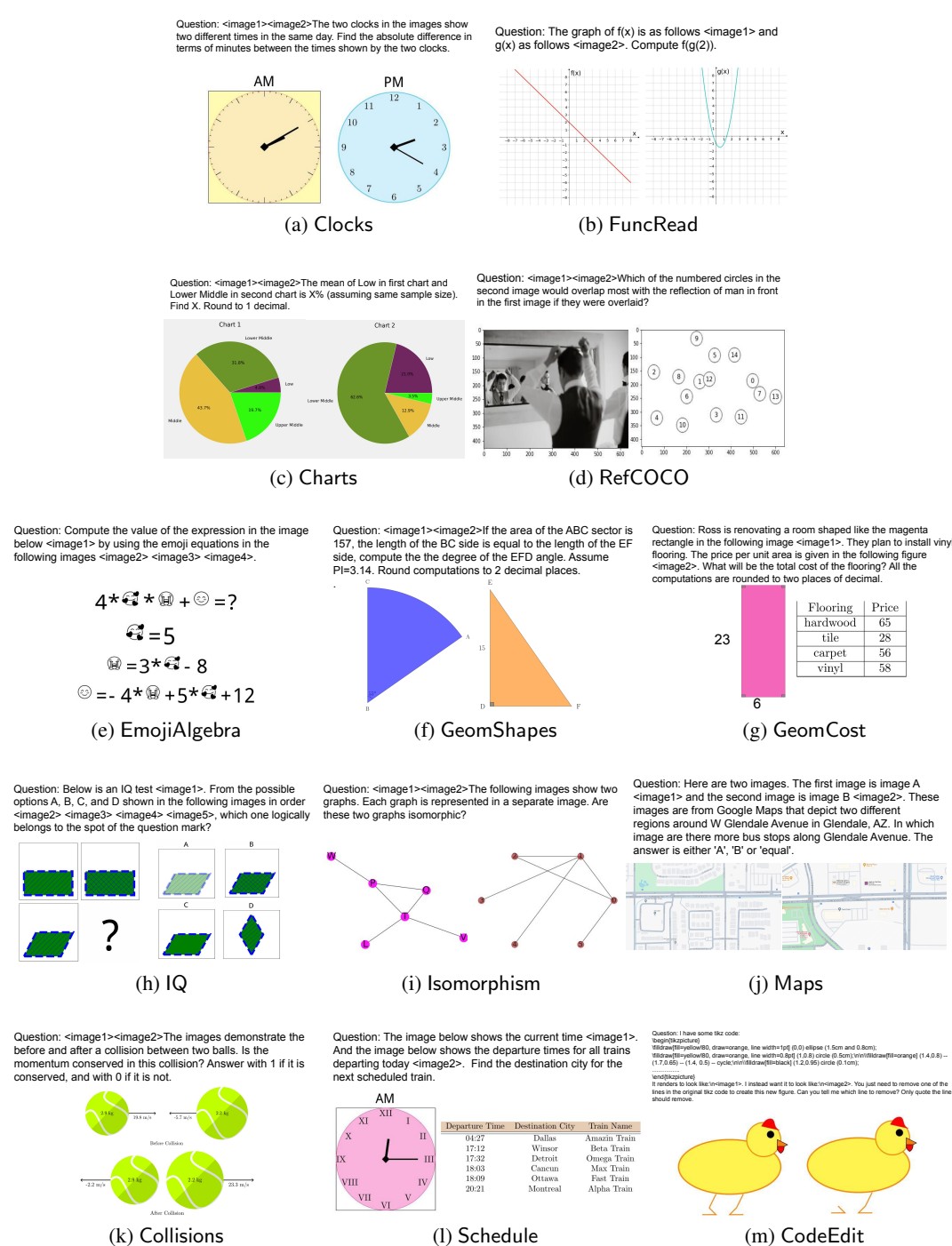

Figure 2: Sample problems from the tasks in ReMI. Note that the images are provided to the models separately in place of the <image i> markers.

image), compute the time difference between them. (7) Schedule: Given the current time (in one image) and a table of train schedules (in another image), answer questions about the next scheduled train. (8) Charts: Given two charts (each in a separate image), possibly in different formats – e.g., one bar chart and one pie chart, identify the differences between the reported values or reason jointly from values in both charts. (9) CodeEdit: Given a TikZ code, the rendered image, and the goal image, determine which line of code should be removed to get to the goal image. (10) Isomorphism:

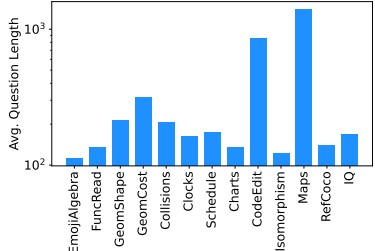

(a) Average length of the questions for each of the tasks in ReMI.

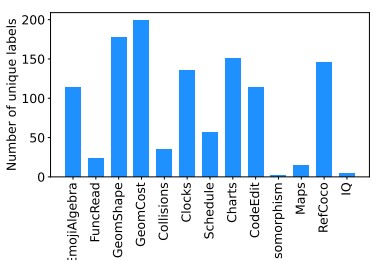

(b) Number of unique labels in each task (for 200 examples).

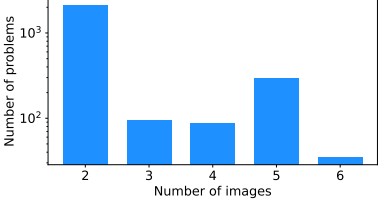

(c) Number of problems in ReMI that have $n$ images, for different values of $n$.

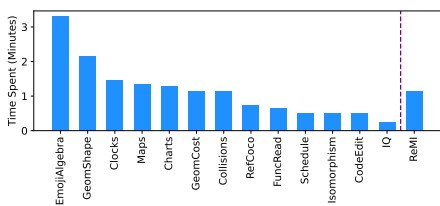

(d) Average time spent on each problem by humans for the human performance results.

Figure 3: Statistics for ReMI and its tasks.

Given two graphs (in two images), determine if they are isomorphic or not. (11) Maps: Given a description of a navigation and four navigation routes on a map (each in a different image), determine which one corresponds to the one in the description. (12) RefCOCO: Given a real-world image and another image of same dimensions with non-overlapping circles marked on it, determine which circle overlaps the most with a target entity in the real image. (13) IQ: Given a matrix of shapes that have a logical connection and with one missing value, predict the shape that goes into the missing part.

Figure 3 presents some extra statistics for the tasks in ReMI, including the average length of the questions per task, the number of unique labels per task, the number of problems with $n$ images for different values of $n$, and the average time spent by humans for solving problems from each task.

## 4 Experiments

We report the performance of multiple state-of-the-art models on our benchmark.

**Metrics:** We mainly report accuracy for our tasks. For textual outputs, we compute exact match while handling slight variations such as spacing issues, lowercase vs uppercase, etc. For numeric answers, we compute a relaxed accuracy with 1% tolerance, mainly to avoid penalizing rounding errors. In the case of relaxed accuracy with tolerance $\epsilon$, a numeric prediction $p$ is considered correct if $(1 - \epsilon)l \leq p \leq (1 + \epsilon)l$ where $l$ is the label. Following the original GeomVerse paper [20], we report relaxed accuracy with 3% tolerance for our GeomShapes and GeomCost tasks as intermediate operations are also rounded and different operation orders lead to slight variations in the final result. For the Clocks, we allow 10 minutes tolerance to account for slight variations in reading times from analog clocks. In our analyses, we also use a metric named *error reduction percentage(ERP)* with respect to a baseline, which corresponds to how much a model reduces the error with respect to a baseline. We define ERP for a baseline as follows:

$$ERP_T(baseline, model) = 100 * \frac{\text{baseline error on task T} - \text{model error on task T}}{\text{baseline error on task T}}$$

Conceptually, the numerator corresponds to how much of the error has been reduced compared to the baseline, and the denominator normalizes by how much room for error reduction existed.

**Naive Baseline:** We provide the expected accuracy for a naive baseline that predicts the answers without looking at the images, by only guessing the final answer based on the text of the question. For

Table 2: The performance of SoTA models on ReMI and its individual tasks. The winner for each task is in bold and the second winner is underlined.

| Task Name | Naive Baseline | Idefics2 | Claude3 Sonnet | Gemini Ultra | Gemini Flash | Gemini 1.5 | GPT4 Turbo | Human |
|---|---|---|---|---|---|---|---|---|
| EmojiAlgebra | 0.0 | 1.0 | 28.0 | 2.5 | 15.0 | 44.5 | **57.5** | 100.0 |
| FuncRead | 5.5 | 11.0 | 24.0 | 15.0 | 36.0 | **40.0** | 26.0 | 100.0 |
| GeomShapes | 0.0 | 14.0 | 17.5 | 14.5 | 34.0 | **51.5** | 32.5 | 100.0 |
| GeomCost | 0.0 | 2.0 | 58.5 | 47.0 | 75.0 | **81.5** | 70.5 | 90.0 |
| Collisions | 30.8 | 31.5 | 51.5 | 36.5 | 56.5 | 50.5 | **62.0** | 100.0 |
| Clocks | 2.0 | 3.0 | **5.0** | 4.0 | 4.0 | 2.5 | 4.0 | 80.0 |
| Schedule | 0.0 | 21.5 | 36.0 | 33.0 | 43.0 | 40.5 | **49.5** | 90.0 |
| Charts | 2.5 | 1.0 | 40.0 | 30.0 | 53.0 | **54.0** | 44.0 | 95.0 |
| CodeEdit | 14.9 | 12.5 | 20.0 | 24.5 | **46.0** | 41.0 | 42.0 | 95.0 |
| Isomorphism | 50.0 | 35.5 | 57.0 | 65.0 | 67.0 | **72.0** | 71.5 | 100.0 |
| Maps | 28.0 | 38.0 | 39.5 | 39.0 | **47.0** | **47.0** | 36.5 | 100.0 |
| RefCOCO | 12.0 | 14.5 | 30.0 | 31.0 | 49.0 | **56.0** | 37.5 | 95.0 |
| IQ | 25.0 | 19.0 | 50.5 | 30.0 | 53.0 | **76.0** | 62.5 | 100.0 |
| ReMI | 13.1 | 15.7 | 35.2 | 28.6 | 44.5 | **50.5** | 45.8 | 95.8 |

multi-choice questions, we assume this baseline will predict the answer correctly with $1/c$ chance where $c$ is the number of choices (for CodeEdit, we consider any line ending in semi-colon to be one of possible choices); for Charts, for every question asking about which cell changed, we assume this baseline responds with $(0, 0)$, and for every question about the number of cells that changed, we assume this baseline responds with $1$; for Clocks, when asking about the difference in time, we assume this baseline always predicts $12 * 60$ minutes; for RefCOCO, we assume this baseline always predicts the circle labeled $0$.

**Models:** We experiment with four state of the art model families, namely Idefics [24], Gemini [46, 38], Claude 3 [5], and GPT4 [35]. From the Idefics family, we experiment with Idefics2. From the Gemini family, we experiment with three models with different sizes and properties, namely Gemini Ultra, Gemini 1.5 Pro, and Gemini Flash. From the Claude 3 family, we experiment with the Sonnet model, and from the GPT4 family, we experiment with GPT4 Turbo. We ran Idefics2 on a single GPU (we used the float16 case without image splitting to ensure the model fits in one GPU) and for the other models we used their APIs.

**Human Performance:** For each task we sampled 20 examples from the test set and had them solved by someone knowledgeable (but not necessarily expert) in that area. We also asked them to measure the amount of time they spent on solving the 20 problems. The average time per problem for each task is reported in Figure 3(d). We observe that some tasks have been more time consuming than the others with EmojiAlgebra being the most time consuming and the IQ being the least time consuming.

### 4.1 Human Baseline Substantially Beats SoTA Models in Multi-Image Reasoning

In Table 2, we present the results of the models as well as the naive baseline and the human performance on the tasks in ReMI. We make the following observations from the obtained results. Firstly, all the models significantly outperform the naive baseline, almost on any task; however, their performance remains far behind the human performance in general, and also in most of the tasks. Secondly, there are some tasks where none of the current models are good at, including Clocks and Isomorphism, where the performances remain quite low[1]. This reveals a potential capability gap in the current state-of-the-art models. Thirdly, we observe that different models perform well on different tasks. For example, Gemini 1.5 substantially outperforms GPT4-Turbo on the IQ, whereas GPT4-Turbo substantially outperforms Gemini 1.5 on EmojiAlgebra. This hints that the frontier models may have different capabilities and limitations.

Hereafter, unless stated otherwise, we do the rest of the experiments with Gemini Pro 1.5, the best overall performing model on ReMI.

---

[1]In the case of the Isomorphism, the dataset is imbalanced with a majority class accuracy of 67 percent.

## 4.2 Single-Image vs Multi-Image Reasoning

We measure whether models perform better when we provide the multiple images separately or when we put them all in a single image and feed them to the model. To this end, we report $ERP_T$(single-image model, multi-image model) corresponding to how much the multi-image model reduces the error with respect to the single-image model for each task $T$. The results are provided in Figure 4. We observe that for most of the tasks, feeding images separately results in positive gains (positive ERP) compared to a single-image case. A manual analysis of the model outputs in the two settings shows that the model may even employ different strategies for solving the problem in these settings. For example, in the case of EmojiAlgebra, we observe that in the single-image case, the model mostly starts by assigning a variable (e.g., $a$, $b$, etc.) to each emoji and then solving the problem by using those variables; However, in the case of multi-image, the model mostly uses either the emojis themselves or their names when doing the calculations.

**Interleaved tasks are affected more:** Out of the six tasks that are positively affected the most (FuncRead, IQ, CodeEdit, Schedule, Isomorphism, and RefCOCO), we observe that five of them (the first five) are interleaved tasks. Averaging the ERP for the interleaved and non-interleaved datasets, we observe a gain of $19.8\%$ for the former case and a gain of $4.9\%$ for the latter case. This hints that reasoning with multiple images might be easier for the models than feeding all images in one image, especially when the images are provided interleaved with text at the right positions.

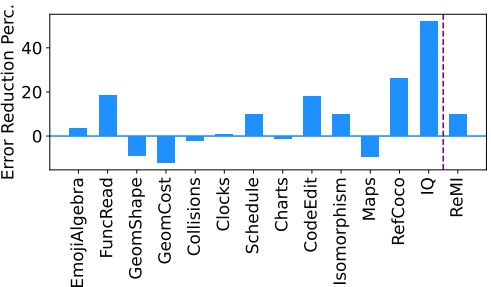

Figure 4: ERP when the images are provided to Gemini 1.5 as multiple vs a single image.

Table 3: Major error sources for each task, identified with a manual inspection of 20 failed examples.

| Task Name | Major Source(s) of Error |
|---|---|
| EmojiAlgebra | 1- Calculation errors, 2- Confusing similar emojis, 3- Misreading from the images (especially minus signs). |
| FuncRead | 1- Model value readings are typically off by about 1 unit. |
| GeomShapes & GeomCost | 1- Calculation errors, 2- Going on a wrong solution path (e.g., computing irrelevant unknown values), 3- Misreading and mis-assigning values (e.g., assigning a length value to the height), 4- Hallucinating non-existent values. |
| Collisions | 1- Not recognizing when two objects are moving together after a collision, 2-Ignoring the velocity vector direction when calculating absolute velocity difference between two objects. |
| Clocks | 1- Not able to read the time properly, 2- Mistaking the minute hand for the hour hand, 3- Not paying attention to the prompt specifying the times are in the same day. |
| Schedule | 1- Not able to read the time properly, 2- Retrieving the wrong values from the table given the time, 3- Sometimes ignoring the 24h format. |
| Charts | 1- Mis-assigning values to the correct row/column in heatmap charts, 2- Under-counting the number of differences in two charts, 3- Reasoning errors of the type *The value decreased from X to X*. |
| CodeEdit | 1-Suggesting removal of nonexistent parts of code, 2-Not properly understanding what each line of code represents in the compiled image. |
| Isomorphism | 1- Jumping prematurely to a conclusion after finding one or two nodes that map to each other, 2- Hallucinating non-existent nodes or edges. |
| Maps | 1- Incorrectly counts similar pins as the same type of objects (e.g., pins that share the same color but different icon), 2- Not paying attention to the prompt specifying the certain area of interest, 3- Gives arbitrary directions that don't match the situation shown in the grid map. |
| RefCOCO | 1- For the image with the circles, coordinate reading is off by ~100-200, 2- Not a proper understanding of spatial clues such as *top right*. |
| IQ | 1- Unfaithful-ness to model's own CoT (e.g., it explains the color should be green, but selects the red), 2- Overly predicting the operation to be rotation (despite no rotation being in the dataset). |

## 4.3 Failure Analysis

For each task, we manually examined 20 examples where the answers given by overall best performing model (Gemini 1.5 Pro) was incorrect and analyzed the dominant reasons behind the failures. This analysis revealed several interesting failure modes – some intuitive and some not – as described below and summarized in Table 3. The diversity of errors observed highlights that this multi-image reasoning domain elicits a wide range of different behaviors that can go wrong in a range of different ways, and that our benchmark tests this wide range of abilities. Calculation errors were present in many of the math-related datasets, so we do not discuss them separately for each task.

For EmojiAlgebra, the overall reasoning process of the model is mostly correct. However, the model sometimes confuses similar emojis. As an example, it assigns a similar (or the same) name to 😂 and 🤣 or to 😍 and 💖 and then these variables get confused in the later calculations. We also observe some misreading of the expressions.

For both Clocks and Schedule, the model suffers from not being able to read the time correctly; e.g. often the minute hand was mistaken for the hour hand. Figure 5 shows a sample clock and times read by the various models. Despite reading the wrong times, the model generally does a good job of computing the time difference given these wrong times, though it sometimes ignores the prompt instructing it to consider both times to be on the same day. In the case of the Schedule, the value retrieved from the table is often not the right value, even given the wrong time read by the mode; Sometimes, this is due the model confusing AM vs PM.

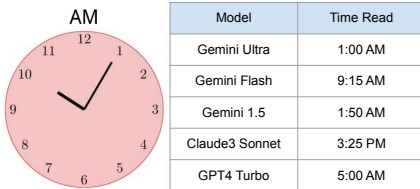

| Model | Time Read |
|---|---|
| Gemini Ultra | 1:00 AM |
| Gemini Flash | 9:15 AM |
| Gemini 1.5 | 1:50 AM |
| Claude3 Sonnet | 3:25 PM |
| GPT4 Turbo | 5:00 AM |

Figure 5: Time read by different models for one of the clocks in the Clocks.

For GeomShapes and GeomCost, the model mostly understands the high-level task (extracting a value from the first shape and using it in the next) and executes it correctly. But it makes reasoning errors on the geometry side where it tries to compute the values for unknown sides/angles that are irrelevant to the question. We also observe some misreading of values or mis-assigning the value for one element to another element (e.g., assigning a side value to a height). Hallucinating non-existent values is another issue. In both cases, the model performs well in understanding and executing the high-level task of extracting a value from the first shape and then using it in the next shape.

For Isomorphism, the model tended to jump to conclusions prematurely, based on some initial guesses. For example, it found one or two nodes that had similar structures and jumped to the conclusion that the graphs are isomorphic, whereas other nodes had different structures. The model also suffered from hallucinating non-existent nodes and edges.

For RefCOCO, the model understands how to use the provided coordinates; however, the coordinates it reads for the circles tends to be off by 10-20%. Moreover, sometimes the model correctly explained that the object of interest is, e.g., on the *top left* but then selected a circle that was not on the *top left*, showing a potential gap in truly understanding what *top left* or other spatial clues are.

For IQ, the model was sometimes unfaithful to its own reasoning (e.g., it explained that the answer must be a green shape, but selected a red shape as the final answer). Also, even though we had no rotation operations in the dataset, the model tended to over-predict the logical operation being rotation, probably due to a prior bias on the presence of rotation in IQ questions.

For FuncRead, the model understands the general logic and follows the calculations correctly, but it fails to correctly read values from the function graphs; the values are mostly off by about 1 unit showing the model can locate the vicinity of the point, but lacks precision.

For Collisions, the model demonstrates issues in interpreting physics diagrams and calculations, particularly in differentiating between elastic and inelastic collisions. It struggles to account for implicit information such as orientation component of the objects velocity.

For Charts, the model reads the correct values from heatmaps, but assigns values to the wrong row/column (typically off by one row/column). Moreover, when we ask the model to identify how many differences there are between two charts, it mostly under-counts. We also see multiple cases where the model claimed a value decreased from $X$ to $X$ (i.e. to the same amount).

For CodeEdit, while correctly identifying the visual changes in the rendered image, the model lacks understanding of how each line of code contributes to the final image. In some cases incorrectly suggests removing code segments that are not present in the original code. Despite these flaws, the model demonstrates some understanding of the code structure, as it avoids suggesting the removal of critical code components that would prevent code from compiling.

For Maps, the model has difficulty counting objects of interest accurately, especially when there are many distractions on the map. It sometimes hallucinates information about restaurants and bars or lists those outside the area of interest. Also, it struggles to differentiate between similar pins, such as coffee shops, bars, and restaurants. When asked about directions, the model's suggestions are often random. While it may list correct streets, the directions it describes do not match the map.

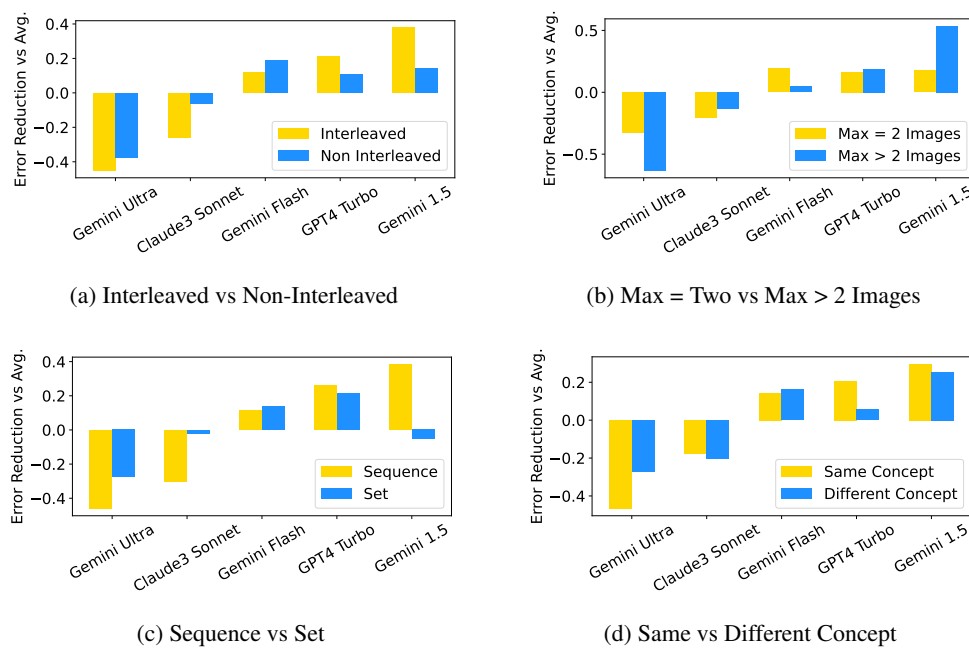

(a) Interleaved vs Non-Interleaved

(b) Max = Two vs Max > 2 Images

(c) Sequence vs Set

(d) Same vs Different Concept

Figure 6: Model performances as a function of the task properties presented in Table 1.

In Figures 9, 10, 11, 12, 13, 14, 15 16 and 17 we present some examples of model failures.

**Reasoning Errors vs Image Reading Errors:** Besides computation errors, we observed that reasoning errors and image reading errors are two of the most dominant sources of failures across the tasks in ReMI. We examined 125 failed examples and verified whether there existed a reasoning error or image reading error in them. The results are provided in Figure 7.

We observe that in 12% of the cases, the values were read correctly from the image and the reasoning was also sound; the failures in these cases were primarily due to minor calculation errors suggesting that while the model understood the problem and approached it correctly, it stumbled in the final execution. In 37.6% of the cases, the image values were read correctly, but the reasoning was incorrect, This is the most frequent error type, indicating that correct reasoning still remains one of the critical gaps even in the state-of-the-art models. In 24.8% of the cases, the model misread some information from the images, but the reasoning is sound. That is, had the model extracted

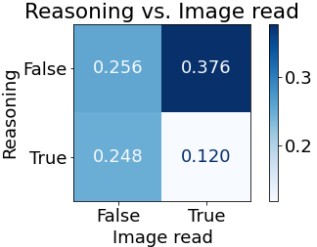

Figure 7: The confusion matrix of reasoning errors vs image reading errors.

the correct information, the final answer could have been correct. This result indicates a second gap in terms of extracting and parsing the correct values from the images and assigning them to the correct components. Finally, in 25.6% of the cases, the model struggled both in extracting information from the image and in applying correct reasoning.

## 4.4 Performance as a Function of Task Properties

In Table 1, we identified multiple distinguishing factors for each of the tasks in ReMI. Here, we aim to measure and compare model performances for tasks exhibiting each property. We note that a naive averaging of a model's performance for datasets in each category and comparing to the other category may be flawed due to: 1- Performances on some tasks being generally higher due to the label space being binary or categorical, and 2- some tasks being generally easier/harder than the other tasks.

To account for the first issue mentioned above, for each model $M$ and task $T$ we compute $P_{M,T}$ as $ERP_T(naive, M)$, i.e. the model's error reduction percentage compared to the naive baseline. This corresponds to how much of the error has been reduced by the model when accounting for random guess, normalized by how much room for error reduction existed when accounting for random guess. To account for the second issue, as a proxy for the hardness of the tasks, we use the average performance of our models on each task $P_T = \frac{\sum_M P_{M,T}}{\text{number of models}}$. We then compute the relative gain compared to the average as $P'_{M,T} = \frac{P_{M,T} - P_T}{P_T}$. Conceptually, this corresponds to the following: After accounting for random noise, how much each model reduced the error with respect to the model-average baseline, on each task. For each model and each group of tasks $\tau$ (e.g., all interleaved tasks), we compute and report the average $\frac{\sum_{T \in \tau} P'_{M,T}}{|\tau|}$.

For this experiment, we excluded Idefics2 due to its poor performance. The results for the other models are reported in Figure 6. According to Figure 6(a), GPT4 Turbo and Gemini 1.5 (the two best performing models) outperform other models on interleaved tasks more than non-interleaved tasks, showing the progress in the frontier models for this recently emerged capability. Figure 6(b) compares the tasks that have a maximum of two images to the tasks where the maximum number of images is more than two. We observe a similar behavior as the interleaved vs non-interleaved case, with Gemini 1.5 gains more on the latter tasks. GPT4 Turbo, however, gains equally on both cases. Interestingly, we observe that while Gemini Flash remains competitive on the former tasks, its performance falls behind on the latter group. According to Figure 6(c), for sequence vs set inputs, we see a stark difference for Claude3 Sonnet and Gemini 1.5. Claude3 Sonnet performs better on set type tasks and Gemini 1.5 performs better on sequence type tasks, but almost loses its advantage on set type tasks. Finally, Figure 6(d) shows that when provided with images corresponding to different concepts, most models show a similar behaviour except for Gemini Ultra that performs better when the concepts are different and GPT4 Turbo that performs better when the concepts are the the same.

## 4.5 Zeroshot vs Fewshot Performance

So far, we examined the performance of various models in a zero-shot setting. We now examine how much of the gap between the model performance and the human performance can be closed by providing fewshot examples as demonstration to the model. Specifically, we prepend two examples along with their manually-written chain of thought solutions to the prompt. We then measured and report $ERP_T(zeroshot, fewshot)$ corresponding the how much the fewshot model reduced the error compared to the zeroshot model. The results are reported in Figure 8.

According to the results, we observe that the overall performance of the model on ReMI improves from $51.5\%$ to $57.9\%$ corresponding to almost $12.5\%$ relative improvement. This shows that LLMs may be capable of learning multi-image reasoning tasks in context and improve their performance. However, the overall performance still remains significantly behind the human baseline which is $95.8\%$. We also see that the amount of improvement is task dependent with some tasks gaining from fewshot examples substantially more than the others.

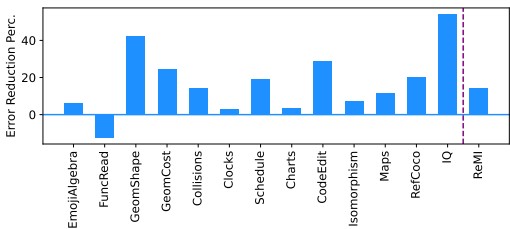

Figure 8: ERP(0-shot, 2-shot) for Gemini 1.5.

## 5 Conclusion

We introduced ReMI, a dedicated benchmark for multi-image reasoning that covers several domains and several key properties that arise when reasoning with multiple images. We evaluated the frontier LLMs on ReMI and compared their performance to humans. The results show a stark gap between model performance and human performance showing a significant room for improvement in the reasoning capabilities of the current state-of-the-art LLMs. Future work can focus on improving LLMs for the limitations found in our failure analysis and measure how much they translate to improvements on ReMI.

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

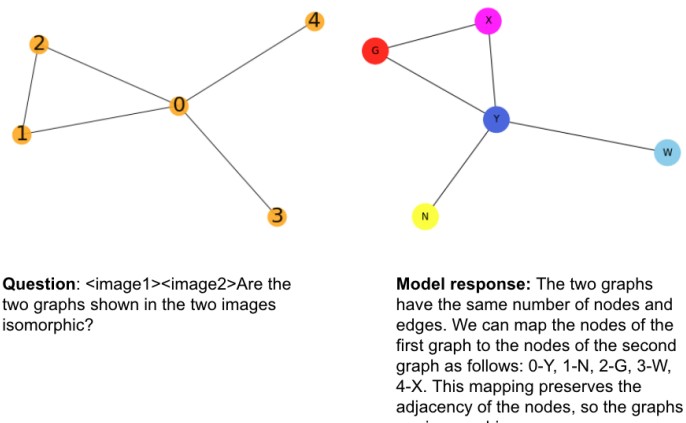

**Question**: <image1><image2>Are the two graphs shown in the two images isomorphic?

**Model response:** The two graphs have the same number of nodes and edges. We can map the nodes of the first graph to the nodes of the second graph as follows: 0-Y, 1-N, 2-G, 3-W, 4-X. This mapping preserves the adjacency of the nodes, so the graphs are isomorphic.

Figure 9: Model failure: Although the model arrives at the correct answer, the mapping showing the isomorphism between the two graphs is wrong.

| Departure Time | Destination City | Train Name |
|---|---|---|
| 00:33 | Mexico City | Max Train |
| 02:31 | Boston | Gamma Train |
| 07:54 | Seattle | Amazin Train |
| 08:34 | Detroit | Omega Train |
| 19:45 | Cancun | Delta Train |

**Question:** The image below shows the current time <image1>. And the image below shows the departure times for all trains departing today <image2>. Find the train name for the next scheduled train.

**Model Response:** "The current time is 9:30 AM. The next scheduled train departs at 09:34 AM. The train name is Omega Train.", "answer": "Omega Train"

Figure 10: Model failure: The model mixes the minutes and hours hand on the clock and also reads 08:34 as 9:34 from the table.

# A Details About the Tasks in ReMI

Below, we provide a detailed description of how each task in ReMI has been created.

- EmojiAlgebra: We created random systems of linear equation where the values for the variables can be derived one-by-one by looking at the equation for which the value for all variables on the right-hand side is known. We also created a random expression with those variables whose value was to be computed. We then created images by replacing the variables with emojis.

- FuncRead: To create this task, we sampled polynomial functions of degree 1, 2, or 3 and plotted their graphs using the matplotlib library. Then we ask the following questions about them: reading values from different functions and summing or subtracting them, computing the limit of a function that is defined as one of the graphs for some domain of values and the other graph for the values outside that domain, function composition, finding a value of interest (e.g., where the derivative is zero) from one graph and reading the other function value at that point, and finding the graph that corresponds to a given function.

- GeomShapes: We generated this dataset by sampling a shape from a set of pre-defined shapes. Each shape has fixed number of pre-defined formulas associated with it corresponding to area, perimeter, angles etc. For each formula, we have input elements and output elements. We first sample one shape and its formula and assign values to input elements, the output element of this formula would be shared with another shape. We then sample another shape and formula whose atleast one input element say $x$ is of same type as output element of the first shape. We assign this element $x$ from the computation of first shape but hide it in the question. We then proceed

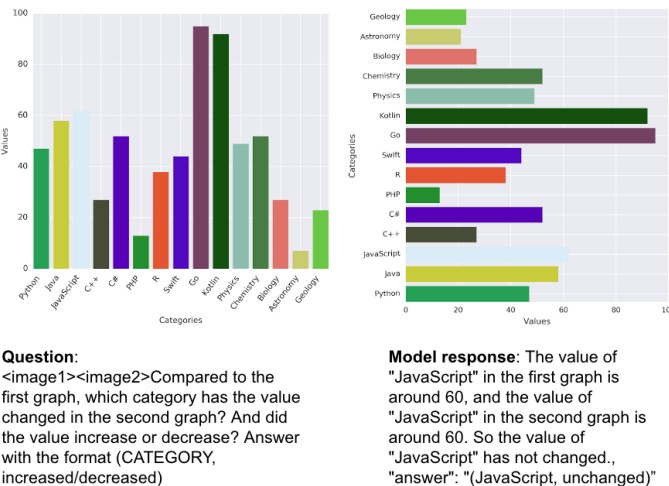

**Question**:
<image1><image2>Compared to the first graph, which category has the value changed in the second graph? And did the value increase or decrease? Answer with the format (CATEGORY, increased/decreased)

**Model response**: The value of "JavaScript" in the first graph is around 60, and the value of "JavaScript" in the second graph is around 60. So the value of "JavaScript" has not changed., "answer": "(JavaScript, unchanged)"

Figure 11: Model failure: The model fails to recognize the category Astronomy whose value has changed and instead outputs that no value has changed.

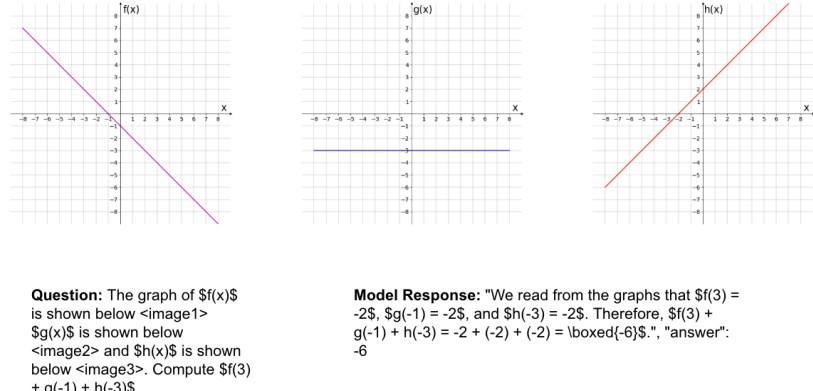

**Question:** The graph of $f(x)$ is shown below <image1> $g(x)$ is shown below <image2> and $h(x)$ is shown below <image3>. Compute $f(3) + g(-1) + h(-3)$.

**Model Response:** "We read from the graphs that $f(3) = -2$, $g(-1) = -2$, and $h(-3) = -2$. Therefore, $f(3) + g(-1) + h(-3) = -2 + (-2) + (-2) = \boxed{-6}$.", "answer": -6

Figure 12: Model failure: The model read's the values wrong from the graph.

to ask the question based on this output formula of second shape. The two questions share this element $x$ which is indicated in the question.

- GeomCost: We generated this dataset by sampling a shape from a pre-selected set of shapes like triangle, parallelogram, square, rectangle etc and selecting one formula out of perimeter and area corresponding to this shape and assigned all the values corresponding to the perimeter or area value. We then choose a template story correspondinf to fencing a boundary, icing the cake etc. out of 10 pre-defined template texts and choose a table corresponding to this template. The table designs are also varied slightly out of fixed number of styles. The cost values are assigned randomly from 1-100.

- Collisions: We created visualizations of two-object collisions, varying initial positions (horizontal, vertical, angled) and randomly assigning masses and velocities. For each collision pair, we then assessed elasticity, coefficient of restitution, and conservation of kinetic energy and momentum.

- Clocks: We generated clock images with different shape, color, style, number representations, etc. using tikz code. Each clock shows a random time and a AM or PM is also added randomly to the image as well. Then, for each pair of images, we compute the difference between their times in terms of minutes and use that as the label.

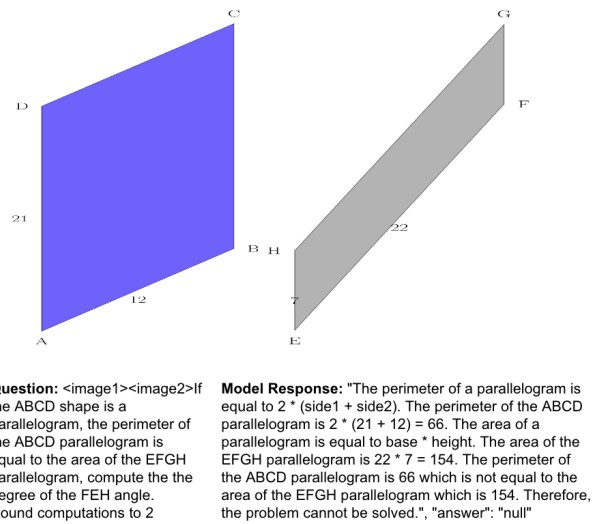

**Question:** <image1><image2>If the ABCD shape is a parallelogram, the perimeter of the ABCD parallelogram is equal to the area of the EFGH parallelogram, compute the the degree of the FEH angle. Round computations to 2 decimal places.

**Model Response:** "The perimeter of a parallelogram is equal to 2 * (side1 + side2). The perimeter of the ABCD parallelogram is 2 * (21 + 12) = 66. The area of a parallelogram is equal to base * height. The area of the EFGH parallelogram is 22 * 7 = 154. The perimeter of the ABCD parallelogram is 66 which is not equal to the area of the EFGH parallelogram which is 154. Therefore, the problem cannot be solved.", "answer": "null"

Figure 13: Model failure: The model wrongly interprets the sides of a parallelogram as its base and height and gets completely off track.

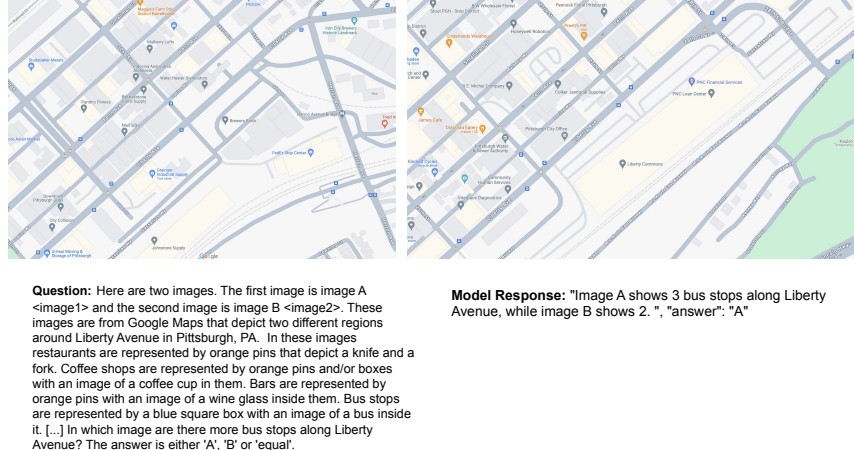

**Question:** Here are two images. The first image is image A <image1> and the second image is image B <image2>. These images are from Google Maps that depict two different regions around Liberty Avenue in Pittsburgh, PA. In these images restaurants are represented by orange pins that depict a knife and a fork. Coffee shops are represented by orange pins and/or boxes with an image of a coffee cup in them. Bars are represented by orange pins with an image of a wine glass inside them. Bus stops are represented by a blue square box with an image of a bus inside it. [...] In which image are there more bus stops along Liberty Avenue? The answer is either 'A', 'B' or 'equal'.

**Model Response:** "Image A shows 3 bus stops along Liberty Avenue, while image B shows 2. ", "answer": "A"

Figure 14: Model failure: The model reads the incorrect number of bus stops along Liberty Avenue in both the images but luckily gets the answer correct.

- Schedule: We generated one clock image showing a random time, similar to the way it was generated for the Clocks. Then, we also generated a random table with different columns (departure time, arrival time, train name, gate, etc.) and with different styles (colors, horizontal/vertical line separators, text rotation, multi-line text, etc.) that included information about the events happening at various times. We then asked questions about the next event happening given the current time shown on the clock.

- Charts: We first randomly generate data matrices and series that are suitable for plotting into four different types of charts: (1) heatmap (2) bar chart (3) line chart (4) pie chart. Then we create a modified version of the data series or matrices by randomly editing one to a few values. This way we obtain pairs of edited data matrices/series. Then we use the Matplotlib library to plot each data matrix/series into a chart by randomly selecting a suitable chart type and randomly choosing a color scheme, layout, etc. for the chart. Heuristics is applied to guarantee that the selected chart type is suitable for plotting the data. Finally we sample from a set of question templates to form QA pairs for each pair of chart. The templates include simple elementary reasoning questions across the two charts or detecting differences of the two charts.

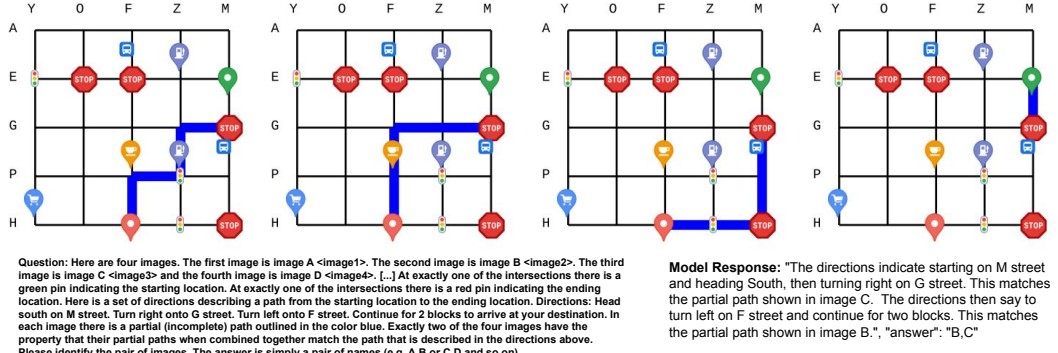

Figure 15: Model failure: The model interprets the wrong image as showing the path described in the text.

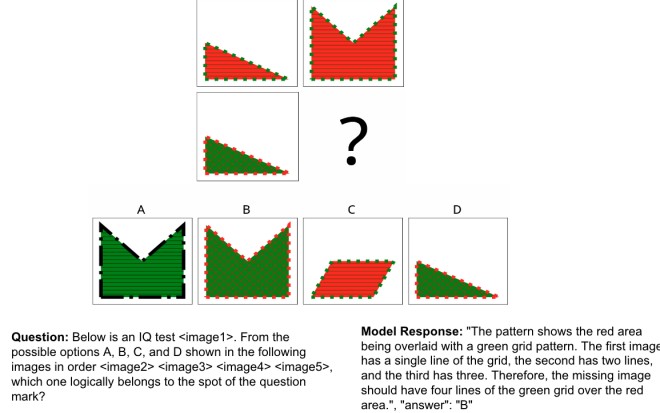

Figure 16: Model failure: The model doesn't seem to understand the pattern properly but guesses the right answer.

- CodeEdit: We first asked a language model to generate tikz code for a list of random objects. We then comment out a single line in the code and recompile it. We only keep the examples where the edited version compiles correctly, and the compiled image is not equal to the original image. A few filters were applied to ensure that the edited image is sensible (e.g. the code being removed is not a variable definition or the beginning of a for loop); specifically, the removed code line had to start with \draw or \filldraw and end with a ;.

- Isomorphism: We used the NetworkX library [14] to generate random graphs using one of the following generators: Erdős-Rényi (ER) graphs [12], scale-free networks (SFN) [7], graphs following the Barabási–Albert (BA) model [4] and stochastic block model (SBM) [17], as well as star, path and complete graphs. Then, for positive examples (i.e. examples where the two graphs are isomorphic), we visualized the same graph with different NetworkX layout, different names for the nodes, and different styles. For the non-isomorphic case, we either sampled two random graphs (this produces easy negative examples) or sampled one random graph and slightly modified it by adding/removing one or two nodes/edges (this produces hard negative examples).

- Maps: Our curated Maps dataset consists of both synthetic and real world examples. We first describe the curation process for the synthetic examples. For synthetic counting queries, we first generate a grid with five horizontal streets and five vertical streets. The street names are randomly assigned in [A..Z]. We then place points of interest (POIs) (gas stations, coffee shops, shopping center and bus stops) at various blocks. We process each block and with a sampling probability of $p = 0.1$ decide whether to place a POI or not. We then pick a POI at random from the list and place it at the block. Similarly, we place traffic lights and stop signs at each

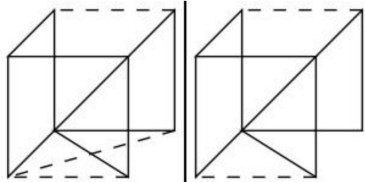

**Question:** I have some tikz code: \documentclass[tikz,border=2pt,png]{standalone} \usepackage{tkz-euclide} \usetkzobj{all} \begin{document} \begin{tikzpicture} \coordinate (O) at (0,0,0); \coordinate (A) at (1,0,0); \coordinate (B) at (1,1,0); \coordinate (C) at (0,1,0); \coordinate (D) at (0,0,1); \coordinate (E) at (1,0,1); \coordinate (F) at (1,1,1); \coordinate (G) at (0,1,1); \draw (O) -- (A) -- (B) -- (F) -- (E) -- cycle; \draw (O) -- (C) -- (G) -- (F) -- cycle; \draw (O) -- (D) -- (G); \draw[dashed] (E) -- (D); \draw[dashed] (B) -- (C); \draw[dashed] (A) -- (D); \end{tikzpicture} \end{document} It renders to look like: <image1> I instead want it to look like: <image2> You just need to remove one of the lines in the original tikz code to create this new figure. Can you tell me which line to remove? Only quote the line I should remove.

**Model Response:** "The line you want to remove is the one that draws the diagonal across the front face of the cube, from point O to point F. This line is unnecessary for visualizing the cube and makes it harder to see the 3D structure.", "answer": "\draw (O) -- (F);"

Figure 17: Model failure: The model fails to correctly match the vertices of the cube to their implicit locations in the image and ends up suggesting to remove the right line of code.

corner with a sampling probability of $p = 0.1$. To generate the second image we copy the above constructed grid and pick at random a particular street. We then pick at random a particular POI on the street and place additional copies of the POI on the street. With a small probability of 0.05 we leave the second image unchanged.

Similarly, for the direction matching queries we generate a grid image as above. We pick a random start and end point and pick a random set of directions between them. We split this direction at a random point to generate two of the four images containing the partial directions. The remaining two images are constructed by picking two different distinct directions at random.

For the case of real data we first prompted a language model to generate a list of 100 cities and an associated street/avenue in that city. We then take this list and for each entry we get two images from Google Maps API that are centered at the particular street. We then manually study the two images and look for distinguishing features (such as bus stops, places of worship, hotels etc.) to construct the query.

- RefCOCO: We sampled 500 imagees from the refCOCO [22] dataset. We then sample 15 points to lie uniformly randomly across the image. We then choose the points that overlap with the goal object as follows. We have the ground truth bounding box of the referred object from the original dataset. We first select the datapoints where at least 1, but less than 8, and include these in `label_inbbox`. We th ar ethe points in the center 25% of the bbob, and so on for provide various precisions for points with 'most overlap': `label_mindist_bboxcenter` is the point that is the closest to the center of the bounding box. `label_25p_tolerance` are the labels in the middle 25% of the bounding box and so on for the `label_50p_tolerance` and `label_75p_tolerance`. Finally we manually check all the datapoints to ensure that the labels points actually overlap with the goal object.

- IQ: We created simple IQ tests where a grid of 2x2 is given as input whose bottom-left value is missing, and four choices are provided as the possible answers from which the model has to select one. The images on the top row are two shapes that are different only in terms of one logical operation. The model has to identify that operation and apply it to the image on the bottom left to find the final answer. We included a number of different shapes (triangles, rectangles, pentagons, parallelograms, etc.) and a number of different logical operations (border color, border pattern, fill color, hatch style, change in shape, etc.). This task is similar in nature to the IQ tasks in [2, 18], but the choices are provided as separate images.

**Quality Check:** To ensure high quality, we went through multiple rounds of checking, where the questions and answers for each task were examined by multiple authors to see any problems can be identified, including whether the label is correct, whether the instructions provided are sufficient to solve the problem and output it in the right format, whether the text of the question is clearly written, whether the images are clearly understandable and the quantities are easily readable, etc. This procedure was done until no more issues could be found for any of the tasks. As a second level of quality check, once we performed our human evaluation, we manually looked into the questions

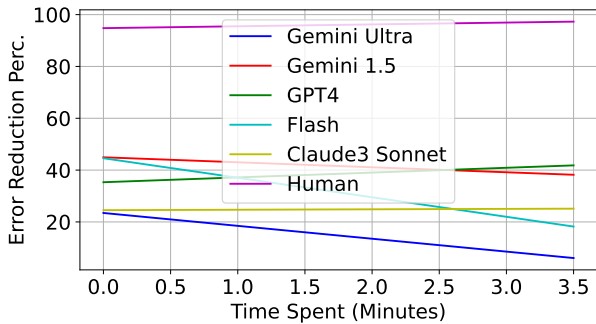

Figure 18: Model gain over naive baseline as a function of the time spent by humans for solving the problems.

where the label provided by the humans disagreed with our labels to ensure that our labels are indeed the correct ones.

## B Performance as a Function of Human Time

In the main text, we reported the average time per problem spent by humans for each task. One may expect that if humans spent more time on a set of problems, those problems might be more difficult for the models. To verify this hypothesis, we fit linear functions to the model performances as a function of time spent by humans and report the results in Figure 18. We observe that only for two of the models (Gemini Ultra and Gemini Flash) the performance goes down as a function of spent time. For other models, the performance almost remains flat.

## C Experimental Setup

For all of the tasks in ReMI, we allowed the models a maximum of 512 output tokens as we observed that when models went beyond that, they were mostly stuck in a wrong path that did not reach a solution and that models could not recover from it. We prompted the model to produce a JSON with two fields: "explanation" containing the step by step reasoning of the model, and the "answer" containing the final answer. We measured the average number of responses that either ended prematurely or did not produce a valid JSON for each model and observed that the numbers were small. Specifically, the numbers for Claude3 Sonnet, Gemini Ultra, Gemini Flash, Gemini 1.5, and GPT4 Turbo were 0.4, 0.3, 0.5, 0.8 and 1.9 percent respectively. For Gemini and Claude, we used the Vertex AI API. For GPT4 Turbo, we used the OpenAI API. For Idefics2, we used the float16 case without image splitting to ensure the model fits in one GPU.

To compute the final performance, we did the following postprocessing on the golden and predicted labels: 1- in the case of string outputs, we lowercased both golden and predicted answers before comparing them, 2- if the predicted label had an extra or missing () around the final answer, we still counted it as true, 3- if the predicted label contained extra units (e.g., producing $20\%$ instead of $20$), we still counted it as true, 4- for the CodeEdit, some lines of codes contained a comment after the code; we considered a predicted label to be true regardless of whether it output the comments or not, 5- we ignored spacing issues and assumed a predicted label to be correct even if it had extra or missing spaces, and finally 6- for the FuncRead, if the golden label was, e.g., $f$ and the predicted label was $f(x)$, we counted it as correct.

## D Limitations

- While our dataset covers a wide range of domains where reasoning over multiple images is required, there may still be many other domains where such reasoning is required that are not covered in our dataset e.g., reasoning about chemicals, reasoning about music sheets, etc.).

- In our experiments for measuring performance as a function of task properties, we had to use proxies to tease apart the effect of random chance and task difficulty. It is possible that with a different procedure for teasing these effects apart, the results change slightly. For this reason, the general patterns observed in those experiments are more important that the small numeric differences.

