# ReMI: A Dataset for Reasoning with Multiple Images — Supplementary Material

**Mehran Kazemi**[1], **Nishanth Dikkala**[2], **Ankit Anand**[1], **Petar Devic**[3],
**Ishita Dasgupta**[1], **Fangyu Liu**[1], **Bahare Fatemi**[2], **Pranjal Awasthi**[2],
**Dee Guo**[2], **Sreenivas Gollapudi**[2], **Ahmed Qureshi**[3]
[1]Google DeepMind, [2]Google Research, [3] Google

## 1 Dataset documentation: datasheet

In this section, we follow the recommendations in Gebru et al. [1] to provide comprehensive documentation for our dataset, available at `https://huggingface.co/datasets/mehrankazemi/ReMI`. The dataset croissant can be found at `https://huggingface.co/api/datasets/mehrankazemi/ReMI/croissant`.

### 1.1 Motivation

- **For what purpose was the dataset created?** Reasoning with multiple images (possible interleaved with text) is an important emerging capability of the LLMs; we created a dedicated dataset to enable tracking progress in this area.

- **Who created the dataset (*e.g.*, which team, research group) and on behalf of which entity (*e.g.*, company, institution, organization)?** The dataset was created by a team of researchers at Google all listed as authors of the paper.

- **Who funded the creation of the dataset?** The dataset was created by Google employees as part of their work at the company, so Google funded its development.

### 1.2 Composition

- **What do the instances that comprise the dataset represent (*e.g.*, documents, photos, people, countries)?** Each instance within the dataset represents a textual question as well as multiple accompanying photos. The photos have a wide range: matplotlib charts and function graphs, NetworkX graphs, latex rendered tables and shapes, screenshots from Google Maps, images from COCO [2], etc.

- **How many instances are there in total (of each type, if appropriate)?** ReMI contains 13 tasks, each having 200 examples for evaluation. All datasets contain photos.

- **Does the dataset contain all possible instances or is it a sample (not necessarily random) of instances from a larger set?** Parts of the dataset have been created programatically. So it is possible to create larger sets. But the released dataset corresponds to the entire data that was used for the paper.

- **What data does each instance consist of?** Each example in the dataset contains a textual reasoning question with image markers representing where the images must be inserted, a list of accompanying images, and the label.

- **Is there a label or target associated with each instance?** Each data point has a label corresponding to the ground truth response to be used for evaluation.

- **Is any information missing from individual instances?** No. All instances have both their text, images and label available.

Submitted to the 38th Conference on Neural Information Processing Systems (NeurIPS 2024) Track on Datasets and Benchmarks. Do not distribute.

- **Are relationships between individual instances made explicit (*e.g.*, users' movie ratings, social network links)?** There is no relationship between different instances.

- **Are there recommended data splits (*e.g.*, training, development/validation, testing)?** This dataset serves as an evaluation benchmark. We have released a test set and a fewshot set.

- **Are there any errors, sources of noise, or redundancies in the dataset?** The dataset has been quality checked over multiple rounds by multiple authors until no errors were found.

- **Is the dataset self-contained, or does it link to or otherwise rely on external resources (e.g., websites, tweets, other datasets)?** The dataset is self-contained and does not link to external resources.

- **Does the dataset contain data that might be considered confidential (e.g., data that is protected by legal privilege or by doctor–patient confidentiality, data that includes the content of individuals' non-public communications)?** The dataset does not contain any confidential or sensitive information.

- **Does the dataset contain data that, if viewed directly, might be offensive, insulting, threatening, or might otherwise cause anxiety?** The dataset does not contain any data that could be considered offensive, insulting, threatening, or anxiety-inducing.

- **Does the dataset identify any subpopulations (e.g., by age, gender)?** The dataset does not identify or contain any information that would allow for the identification of subpopulations based on attributes.

- **Is it possible to identify individuals (i.e., one or more natural persons), either directly or indirectly (i.e., in combination with other data) from the dataset?** The dataset does not contain any information that could be used to directly or indirectly identify individuals, either on its own or in combination with other data.

- **Does the dataset contain data that might be considered sensitive in any way (e.g., data that reveals race or ethnic origins, sexual orientations, religious beliefs, political opinions or union memberships, or locations; financial or health data; biometric or genetic data; forms of government identification, such as social security numbers; criminal history)?** The dataset does not contain any sensitive data that could reveal attributes.

## 1.3 Collection

- **How was the data associated with each instance acquired?** The text of the questions have been created automatically using code, or, in the case of one task, by prompting a language model. The images are acquired from three sources: 1- Automatically generated using visualization libraries, 2- screenshots from Google Maps, and 3- from the COCO dataset.

- **What mechanisms or procedures were used to collect the data (e.g., hardware apparatuses or sensors, manual human curation, software programs, software APIs)?** The data collection process primarily involved a combination of automated procedures and manual human input. The majority of the data collection was performed using software programs and scripts that were developed and executed by the authors. These programs included algorithms and techniques designed to generate and curate the specific types of data required for the dataset.

- **If the dataset is a sample from a larger set, what was the sampling strategy (e.g., deterministic, probabilistic with specific sampling probabilities)?** NA

- **Who was involved in the data collection process (e.g., students, crowdworkers, contractors) and how were they compensated (e.g., how much were crowdworkers paid)?** The data collection process was conducted solely by the authors of the paper.

- **Over what timeframe was the data collected?** The data collection process took place over a four-month period.

- **Were any ethical review processes conducted (e.g., by an institutional review board)?** No.

- **Did you collect the data from the individuals in question directly, or obtain it via third parties or other sources (e.g., websites)?** We collected data from no individuals.

- **Were the individuals in question notified about the data collection?** NA.

- **Did the individuals in question consent to the collection and use of their data?** NA.

- **If consent was obtained, were the consenting individuals provided with a mechanism to revoke their consent in the future or for certain uses?** NA.

- **Has an analysis of the potential impact of the dataset and its use on data subjects (e.g., a data protection impact analysis) been conducted?** NA.

## 1.4 Uses

- **Has the dataset been used for any tasks already?** The dataset is used for benchmarking model performances on multi-image reasoning, in this paper.

- **Is there a repository that links to any or all papers or systems that use the dataset?** The dataset is generated and used in this paper only.

- **What (other) tasks could the dataset be used for?** To measure model performance on multi-image reasoning.

- **Is there anything about the composition of the dataset or the way it was collected and preprocessed/cleaned/labeled that might impact future uses?** No.

- **Are there tasks for which the dataset should not be used?** The main purpose of the dataset is to be used for evaluation and it should not be used for training.

## 1.5 Distribution

- **Will the dataset be distributed to third parties outside of the entity (e.g., company, institution, organization) on behalf of which the dataset was created?** Yes, the dataset is available publicly in Huggingface.

- **How will the dataset will be distributed (e.g., tarball on website, API, GitHub)?** The dataset is distributed through Huggingface.

- **When will the dataset be distributed?** The dataset is already available.

- **Will the dataset be distributed under a copyright or other intellectual property (IP) license, and/or under applicable terms of use (ToU)?** In the spirit of open science and collaboration, we have released the datasets under a Creative Commons Attribution 4.0 International (CC BY 4.0) license. For comprehensive details about the terms of the CC BY 4.0 license, please visit the Creative Commons website: `https://creativecommons.org/licenses/by/4.0/`.

- **Have any third parties imposed IP-based or other restrictions on the data associated with the instances?** Please refer to the copyright.

- **Do any export controls or other regulatory restrictions apply to the dataset or to individual instances?** Please refer to the copyright.

## 1.6 Maintenance

- **Who will be supporting/hosting/maintaining the dataset?** The authors will support, host, and maintain the dataset.

- **How can the owner/curator/manager of the dataset be contacted (e.g., email address)?** The owner (Mehran Kazemi) can be contacted through `mehrankazemi@google.com`.

- **Is there an erratum?** No. If errors are found in the future, we will release errata using the same link.

- **Will the dataset be updated (e.g., to correct labeling errors, add new instances, delete instances)?** If any issues are found in the dataset, it will be updated to ensure correctness.

- **If the dataset relates to people, are there applicable limits on the retention of the data associated with the instances (e.g., were the individuals in question told that their data would be retained for a fixed period of time and then deleted)?** NA.

- **Will older versions of the dataset continue to be supported/hosted/maintained?** Yes, older versions of the dataset will continue to be maintained and hosted.

- **If others want to extend/augment/build on/contribute to the dataset, is there a mechanism for them to do so?** Yes, others are welcome to extend, augment, build on, or contribute to the dataset. They are encouraged to download the dataset, create their own modified versions, and publish their work on their preferred platform.

## 2   Accessibility

The datasets created in this research are available for download at `https://huggingface.co/datasets/mehrankazemi/ReMI`. In the spirit of open science and collaboration, we have released the datasets under a Creative Commons Attribution 4.0 International (CC BY 4.0) license. For comprehensive details about the terms of the CC BY 4.0 license, please visit the Creative Commons website: `https://creativecommons.org/licenses/by/4.0/`.

## 3   Reproducibility

In our pursuit of reproducibility, we conducted experiments with five prominent models: Gemini Ultra, Gemini 1.5 Pro, Gemini Flash, Claude Sonnet, and GPT4 Turbo. All of these variants are publicly accessible through various platforms, ensuring transparency and enabling further research. For the Gemini and Claude models, we utilized the Google Cloud Platform (Vertex AI) as our computational infrastructure to execute the experiments. For GPT4, we used the OpenAI API.

The author bear all responsibility in case of violation of rights.