# OpenReview forum: "ReMI: A Dataset for Reasoning with Multiple Images"
_NeurIPS.cc/2024/Datasets_and_Benchmarks_Track — NeurIPS 2024 Track Datasets and Benchmarks Poster_

### Official Review · Reviewer_FcqE · 2024-07-15
**A dataset to test MLLMs' ability to reason over multiple images**

**Rating:** 7
**Confidence:** 4
**Correctness:** Yes
**Clarity:** Yes

**Review:**

This paper proposes a ReMI dataset that test MLLM's ability in reasoning over multiple images. The nice thing about this work is the author divide the datasets into several dimensions, such as Algebra, Calculus, Code, Chart and etc, so that the users can use ReMI to evaluate a model's fine-grained abilities. Also, the number of images can span from 2 to 6, so that not only limited to the bi-image setting.

**Strengths:**

With the advances of MLLMs, single image QA is no longer sufficient test a model's ability. Therefore, the contribution of this paper is exactly needed for the next year MLLM research with longer and more complex multimodal context.

**Additional Feedback:**

N/A

**Documentation:**

Yes

**Limitations:**

Yes.

**Opportunities For Improvement:**

The author does not include temporal multiple images (e.g. video frames) in this dataset, which is also a typical use case.

**Relation To Prior Work:**

Yes

**Summary And Contributions:**

he paper introduces ReMI, a new benchmark dataset designed to assess the capability of Large Language Models (LLMs) to perform reasoning with multiple images. The ReMI dataset encompasses a variety of tasks from domains such as math, physics, logic, coding, and spatial and temporal reasoning, which require the integration of information across different modalities. The authors have evaluated several state-of-the-art LLMs using ReMI and found a significant gap between the models' performance and human-level proficiency. The paper also provides an analysis of the models' strengths and weaknesses, highlighting areas for future improvement. To promote further research, the ReMI dataset is being open-sourced.

1. The creation of ReMI, a comprehensive benchmark dataset for evaluating multi-image reasoning abilities of LLMs across diverse domains.
2. An extensive evaluation of several cutting-edge LLMs on the ReMI benchmark, revealing a substantial performance gap compared to human-level proficiency.
3. The open-sourcing of the ReMI dataset to foster continued research and development in multi-image reasoning capabilities of LLMs.

---

> ### Author Rebuttal · Authors · 2024-08-16
>
> We thank the reviewer for their great feedback.
>
> > The author does not include temporal multiple images (e.g. video frames) in this dataset, which is also a typical use case.
>
> We agree that video frames are also a typical use-case of multi-image reasoning, but we found that SEED-Bench-2 provides a good coverage of that use-case, so we decided to mainly focus on other less-covered use-cases.
>
> > Other concerns?
>
> If any other concerns about our work remains, please let us know and we are happy to address them.

---

> > ### Comment · Reviewer_FcqE · 2024-08-29
> >
> > Thank you for your response. My score will stay.

---

### Official Review · Reviewer_Lwd4 · 2024-07-25

**Rating:** 7
**Confidence:** 3
**Correctness:** Yes
**Clarity:** Yes

**Review:**

Strengths:
1. The proposed benchmark for multi-step image reasoning is well motivated and appears to serve a strong test for current LLMs, given the large disparity in the best LLM vs human performance.

2. The paper is quite complete in its description of the entailed tasks, dataset breakdown and construction and analysis of existing models.

3. The paper is comprehensive in analyzing different factors that can influence performances of current models and also includes qualitative analysis (although limited to 20 examples). Overall, the detailed results and breakdown for each section are interesting and well described.

Opportunities for Improvement:
1. Regarding few-shot performance -- currently it appears a 2-shot prompt can by itself lead to ~ 6% absolute performance improvement and 12.5% relative improvement. How does increasing the number of prompt examples beyond 2 influence performance, and does it plateau at a point?  I think this important to identify, as currently it may just be that LLMs may not have learned how to do multi-image reasoning/the selected task through base pretraining itself and hence require multiple in-context examples.

2. How do instruction-tuned vision-language models such as LLAVA or general ones such as BLIP-2 perform on the task? Similarly, how do frameworks such as VisProg and ViperGPT perform on these tasks? Considering these methods as additional baselines to the current LLMs can be useful in identifying benefits and drawbacks of a wider range of models.

**Strengths:**

Please see review

**Additional Feedback:**

n/a

**Documentation:**

It is mentioned dataset will be open-sourced, and dataset creation details are described in appendix.

**Limitations:**

Yes, limitations are discussed

**Opportunities For Improvement:**

Please see review

**Relation To Prior Work:**

Yes

**Summary And Contributions:**

The paper introduces a benchmark for assessing capabilities of current LLMs on multi-image reasoning tasks wherein models are required to reason over 2 to 6 images (depending on the task type). The benchmark covers diverse domains such as algebra, calculus, physics, etc and associated reasoning capabilities. Performances of current LLMs (best being 50.5%) are found to be substantially lower than humans (95.8%). Detailed analysis is provided to i) breakdown performances of current LLMs across different task types (and also on collated common task properties), ii) identify benefits in providing images separately to LLMs as opposed to together as a single-image, iii) identifying error sources of the best performing model (Gemini1.5 Pro) through manual inspection of 20 examples, iv) distinguishing reasoning vs image reading errors, and iv) beneficial impact of few-shot (2-shot) prompting.

---

> ### Author Rebuttal · Authors · 2024-08-16
>
> We thank the reviewer for thoughtful feedback.
>
> > More shots
>
> We evaluated models in a 2-shot setting as we were restricted by the maximum number of images (=16) that Vertex AI accepted as input; we wanted to stick to Vertex AI to ensure reproducibility. To test the reviewer's hypothesis, we evaluated Gemini Flash with in 2-shot and 5-shot settings, on three of our tasks where each example had a maximum of two images. On all three tasks, the model performance decreased when we increased the number of shots from 2 to 5. Specifically, for Collisions, the performance decreased by 9\%, on GeomCost it decreased by 2\%, and on Charts it decreased by 5\%. So while adding a few examples may generally help the model learn the task and perform better, increasing the number of shots seems to have an adverse effect. This might be because increasing the number of shots increases the number of images substantially, making the problem even more multi-image-y.
>
> > Other baselines
>
> In terms of evaluating open-weight models, we would like to start by referring the reviewer to a clause in the LICENSE of the llama model (https://github.com/meta-llama/llama/blob/main/LICENSE) that prohibits us from interacting with the model (see line 65-71 of the license). Since a large number of the existing open-weight models are either derivatives of llama or built on it in one way or another (including some of the ones suggested by the reviewer), we are unfortunately not able to report results for those models. This is an unfortunate and non-ideal situation, but please note that this is Meta's policy and is outside the control of the authors.
>
> We evaluated one of the state-of-the-art open-weight multi-image models, namely idefics2, which has a more permissive license (we used the float16 case without image splitting to ensure the model fits in one GPU). The results are as follows.
>
> | Task         | Accu  |
> |--------------|-------|
> | FuncRead     | 11.0  |
> | Maps         | 38.0  |
> | RefCoco      | 14.5  |
> | GeomCost     | 2.0   |
> | Collisions   | 31.5  |
> | Isomorphism  | 35.0  |
> | Schedule     | 21.5  |
> | GeomShape    | 14.0  |
> | CodeEdit     | 12.5  |
> | IQ           | 19.0  |
> | EmojiAlgebra | 1.0   |
> | Charts       | 1.0   |
> | Clocks       | 3.0   |
> | **Average**  | **15.7** |
>
> Based on the results, we see a clear gap between idefics2 and the other models that were evaluated. Notice that for some tasks, the model performance is even lower than random chance. That is because we noticed several cases where the model produces degenerate text (e.g., keeps repeating the same sentence) in which case no answer will be extracted from the model.
>
> > Other concerns?
>
> If any other concerns remain, please let us know and we are happy to address them.

---

> > ### Comment · Reviewer_Lwd4 · 2024-08-22
> >
> > Thank you for the clarifications and additional results. My concerns are largely addressed and I maintain my accept rating.

---

### Official Review · Reviewer_ZMgP · 2024-07-26
**Interesting setting for benchmarking**

**Rating:** 6
**Confidence:** 3
**Correctness:** Generally yes.
**Clarity:** Yes, the paper is well written and th…

**Review:**

Pros:
- Interesting setting. The complex reasoning with multiple images are not well examined and studied before.
- The picked 13 tasks are insightful and representative for examining the multi-image complex reasoning ability of MLLMs.
- Transparency. The dataset is provided in the submission and the details regarding this dataset are generally disclosed. I appreciate that.
- The writing is generally good and the demonstration is nice.

Cons:
- The small data size. I understand the proposed dataset is evaluation-only, and collecting such multi-image complex reasoning samples are costly. Still, I find 200 samples for each task and 2600 samples in total to be a little too small for producing a trustworthy and accurate result. Besides, such a small size makes it easier to game and cheat for existing MLLMs.
- Lack of evaluation with open-source MLLMs. I understand that recent open-source MLLMs may not be as good as close-source ones, but it will still be beneficial to show how far they are to close-source ones and human baselines.

**Strengths:**

Please see the pros above.

**Additional Feedback:**

Please see the pros and cons above. I'm looking forward to the rebuttal.

**Documentation:**

Yes.

**Ethics:**

No.

**Limitations:**

Please see the cons above.

**Opportunities For Improvement:**

Some may include:

- Larger dataset size.
- Evaluation with open-source models and more corresponding analysis.

Please see the cons above.

**Relation To Prior Work:**

Yes, discussed well.

**Summary And Contributions:**

- This paper focuses on the complex reasoning in multi-image scenario for MLLM.
- A benchmark is provided, covering 13 tasks  of complex reasoning with multiple images.
- Experimental analysis is performed, which shows the benchmark is challenging and existing MLLMs fails behind human baseline largely.

---

> ### Author Rebuttal · Authors · 2024-08-16
>
> We thank the reviewer for the great feedback on our work.
>
> > The small data size.
>
> * We decided to keep a smaller dataset size for two reasons: 1- multi-image reasoning is an expensive task and we wanted to make the dataset available to a wider group of researchers (especially those having access to lower amounts of compute), and 2- we wanted to be able to manually check the problems and ensure high quality for the final dataset.
> * We anticipate that in the future, the dataset may be used mostly in two settings: either evaluating on the entire dataset, or evaluating on subsets of tasks that have a common property (e.g., all interleaved tasks); the individual tasks may be used mostly for failure analysis purposes. In both cases, multiple tasks will be included in the evaluation set comprising a big-enough set.
> * We also note that the presence of 13 diverse tasks in ReMI (with various properties -- see Table 1) guards against gaming and cheating.
>
> > Lack of evaluation with open-source MLLMs.
>
> We evaluated one of the state-of-the-art open-weight multi-image models, namely idefics2. The results are as follows (we used the float16 case without image splitting to ensure the model fits in one GPU).
>
> | Task         | Accu  |
> |--------------|-------|
> | FuncRead     | 11.0  |
> | Maps         | 38.0  |
> | RefCoco      | 14.5  |
> | GeomCost     | 2.0   |
> | Collisions   | 31.5  |
> | Isomorphism  | 35.0  |
> | Schedule     | 21.5  |
> | GeomShape    | 14.0  |
> | CodeEdit     | 12.5  |
> | IQ           | 19.0  |
> | EmojiAlgebra | 1.0   |
> | Charts       | 1.0   |
> | Clocks       | 3.0   |
> | **Average**  | **15.7** |
>
> Based on the results, we see a clear gap between idefics2 and the other models that were evaluated. Notice that for some tasks, the model performance is even lower than random chance (e.g., IQ). That is because we noticed several cases where the model produces degenerate text (e.g., keeps repeating the same sentence) in which case no answer will be extracted from the model.
>
> While we were also keen to evaluate other open-weight models, the license of the llama model (https://github.com/meta-llama/llama/blob/main/LICENSE) prohibited us from interacting with this model (see line 65-71 of the license) and several others that are either derivatives of llama or build on it in one way or another. This is an unfortunate and non-ideal situation, but please note that this is Meta's policy and is outside the control of the authors.
>
> > Other concerns?
>
> If there are any other concerns, please let us know and we are happy to address them.

---

### Official Review · Reviewer_rtaP · 2024-07-28
**ReMI: A Dataset for Reasoning with Multiple Images**

**Rating:** 7
**Confidence:** 3
**Correctness:** Yes
**Clarity:** Yes

**Review:**

The paper exhibits a number of multi-image question answering tasks which highlight the deficiencies of current LLMs. Some failure analysis is provided, as well as a comparison of zeroshot vs. fewshot performance.

**Strengths:**

The dataset covers a diverse range of tasks and tests a number of different modalities. The summary of related work is comprehensive.

**Additional Feedback:**

.

**Documentation:**

Yes

**Limitations:**

Yes

**Opportunities For Improvement:**

The dataset provides only two training examples in each category. This allows for fewshot learning but precludes other approaches such as traditional training and testing, or fine tuning of LLM's on training data. It would be better if a larger amount of training data could be provided, thus allowing other approaches to be explored, analysed and evaluated on the same test data.

**Relation To Prior Work:**

Yes

**Summary And Contributions:**

This paper introduces the Reason with Multiple Images (ReMI) dataset. ReMI aims to test multi-image visual question answering across multiple domains including math, physics, logic, code, table/chart understanding,and spatial/temporal reasoning. It is tested against a naive baseline and three state-of-the-art LLM's as well as human performance.

---

> ### Author Rebuttal · Authors · 2024-08-16
>
> We thank the reviewer for great feedback.
>
> > The dataset provides only two training examples in each category.
>
> Our goal in creating ReMI was to provide a benchmark over which the general multi-image reasoning capability of the models can be measured and tracked over time. Providing a large train set has the downside that future models may train on ReMI-train and show higher numbers on ReMI-test (since it is in-distribution with the train data), but that improvement may not be reflective of actual progress in multi-image reasoning (i.e. models could work arbitrarily well on ReMI-test but still be poor at multi-image reasoning outside of ReMI's distribution). While the way we constructed ReMI allows for generating a larger number of data points and providing them as train set, we intentionally decided to not provide a train set to avoid the above problem.
>
> Note that several recent widely-adopted evaluation datasets indeed come with only test sets. That includes, among others, MathVista (where only a test set is released), MMMU (where only 5 examples per category are released for few-shot, and the rest are for validation/test), MATH-V (where only a test set is released), BLINK (where only validation/test sets are provided), and many other datasets.
>
> > Other concerns?
>
> If any other concerns about our work remains, please let us know and we are happy to address them.

---

### Decision · Program_Chairs · 2024-09-26

**Decision:**

Accept (Poster)

**Comment:**

The ReMi paper introduces a new dataset / evaluation benchmark for evaluating visual reasoning with multiple images. Humans perform well on this baseline while the large, multi-modal, language models studied in this paper perform poorly. The paper is clear and well-written, this area is not well-studied, this dataset is diverse, and the review of related work is comprehensive.

There were two weaknesses consistently identified by the reviewers:
1) The evaluation contains a relatively small number of samples (only ~2600 total, 200 per task). The authors point out this is due to the cost of collecting and annotating such a dataset and that some similar datasets (e.g. MathVista) have similar approaches.
2) SotA OSS models are not evaluated. The authors point out that this is because most SotA OSS models are derived from the LLAMA models from Meta and they are prohibited by the license from using LLAMA or LLAMA derivatives because of the size of their organizations user base (see line 65 https://github.com/meta-llama/llama/blob/main/LICENSE ).

For issue (2), I am not concerned since publishing and releasing this paper / dataset will allow the scientific community to evaluate the LLAMA series of models, so this provides a reason to publish the dataset sooner so that other research institutions can fill in this gap.

For (1), I agree with the reviewers that the small size of the dataset is a weakness for the paper and may mean the results are noisy or may not generalize as compared to a more extensive evaluation set. While a diverse set of task types does mitigate the generalization risk somewhat, a larger evaluation set would increase the usefulness of this benchmark and I still think there is some risk of noisy evaluations on the individual tasks. However, I think the positives of this dataset outweighs this issue and am recommending an accept. I would encourage the authors to consider expanding the benchmark for future work.

Thank you for the submission!

One nit, on line 67 in the intro, the word language is misspelled.